# Evaluating Spatial Understanding of Large Language Models

**Yutaro Yamada** *,♣        **Yihan Bao** *,♣        **Andrew K. Lampinen**
**Jungo Kasai**◇        **Ilker Yildirim**♣

♣ **Yale University**      ◇ **Toyota Technological Institute at Chicago**
{yutaro.yamada, yihan.bao, ilker.yildirim}@yale.edu
Reviewed on OpenReview: https://openreview.net/forum?id=xkiflfKCw3

## Abstract

Large language models (LLMs) show remarkable capabilities across a variety of tasks. Despite the models only seeing text in training, several recent studies suggest that LLM representations implicitly capture aspects of the underlying grounded concepts. Here, we explore LLM representations of a particularly salient kind of grounded knowledge — spatial relationships. We design natural-language navigation tasks and evaluate the ability of LLMs, in particular GPT-3.5-turbo, GPT-4, and Llama2 series models, to represent and reason about spatial structures. These tasks reveal substantial variability in LLM performance across different spatial structures, including square, hexagonal, and triangular grids, rings, and trees. In extensive error analysis, we find that LLMs' mistakes reflect both spatial and non-spatial factors. These findings suggest that LLMs appear to capture certain aspects of spatial structure implicitly, but room for improvement remains.[1]

## 1 Introduction

Large language models (LLMs) show remarkable capabilities in language, and also hints of implicitly learning about the grounded concepts beyond language. For example, language models can develop semantically-organized internal representations for basic concepts like color and direction (Abdou et al., 2021; Patel & Pavlick, 2022) — which can allow grounding the models with only a few examples. Furthermore Li et al. (2021) demonstrate that internal representations of language models can dynamically track the states of entities and their relations during discourse. Human language use manifests the semantics of the world from which it originates, and thereby might allow LLMs to implicitly learn something about the entities and processes that exist in the physical world.

Natural intelligences extract and use such knowledge of the physical world – often referred to as world models. A particularly salient example is the ability of humans and animals to create and manipulate mental maps, which serves as a fundamental prerequisite for flexibly navigating and interacting with their environments. Cognitive maps (Tolman, 1948) were suggested as a metaphor for mental representations that enable adaptable behavior such as planning routes or finding shortcuts. The quest to uncover how the brain represents such maps has led to significant discoveries about the neural mechanisms underlying such maps, such as place cells (O'Keefe & Dostrovsky, 1971), grid cells (Hafting et al., 2005), and boundary cells (Lever et al., 2009). While navigation generally involves active, grounded experience, some studies suggest that humans use similar representational structures for abstract knowledge as well (e.g. Whittington et al., 2020; Constantinescu et al., 2016). Furthermore, cognitive and neural evidence suggests that humans and animals can learn spatial structure solely from sequences of observations (Whittington et al., 2022; Garvert et al., 2017). This raises an intriguing possibility – that LLMs might also be capable of inferring sophisticated spatial relations from their sequential, text-based inputs.

---

* Equal contribution.
[1]Our code and data are available at https://github.com/runopti/SpatialEvalLLM, https://huggingface.co/datasets/yyamada/SpatialEvalLLM

In this paper, we examine the spatial understanding capabilities of LLMs – in particular OpenAI's GPT-3.5-turbo, GPT-4 models as well as Llama2-7B, Llama2-13B, Llama2-70B, CodeLlama-34B models (Touvron et al., 2023). We designed a broad set of navigation tasks rendered in natural language such that successfully solving these tasks requires accurately representing the underlying spatial relations. These relations include grids with square, hexagonal, and triangular topologies, in addition to rings and trees. Our study reveals that LLMs exhibit varying performance when the underlying spatial structures differ (§3.1). We also observe that presenting the global map upfront actually makes the task more challenging compared to providing local navigational instructions only (§3.2). Moreover, we investigate the effect of the spatial patterns (e.g., random order, row-major) by which global map is expressed on the performance of LLMs (§3.3). We also provide evidence that LLMs spontaneously utilize object information as landmarks for constructing spatial maps (§3.5), much like humans and animals. Finally, detailed error analyses (§4) confirm that in spatial structures that LLMs perform well, their mistakes manifest the underlying spatial topology, as well as non-spatial factors. These error distributions suggest that GPT-4, which often substantially outperforms GPT-3.5, seems to implicitly grasp certain elements of spatial structure, but there is still room for improvement.

We believe gaining insights into the spatial comprehension abilities of LLMs is valuable in enhancing our understanding of how these models acquire and grasp grounded concepts.

## 2 Spatial Understanding Task

How can we evaluate the text-only models' understanding of spatial information? With human participants, we could have them explore the environment and then ask them to draw a map. However, for text-in, text-out models like LLMs, formalizing the task of spatial reasoning is difficult because these models lack the capability to directly interact with the physical world or visually draw the entire map. However, some studies in human cognition (Garvert et al., 2017) have presented participants with sequential data that is sampled from an underlying spatial structure. These studies suggest that humans implicitly acquire knowledge and learn representations that mirror the spatial structure that is latent in the data. This motivates the hypothesis that presenting sequential transitions might be enough for LLMs to achieve spatial understanding.

For example, if a model comprehends a square map's structure, it should be able to answer the following question: "You start at a spot where you find an apple. You move up and find a banana. Then you move right and find an orange. Next, you move down and find a grape. Now, you move left. What do you find?" Answering this question correctly demonstrates an understanding of loop closure, which is a fundamental aspect of this spatial structure. That is, if we have a square grid and an initial location, we can allow the model to take a random walk until it reaches a location it has already visited before. At each newly visited location, we inform the model about the objects it perceives, and then we ask the model which object it would have seen just before reaching the already visited location. By generating such questions synthetically, we can systematically evaluate the spatial understanding of LLMs. For each question, we randomly select the object names from the ImageNet-1k labels to fill every location of the spatial grid to create the underlying map.

### 2.1 Models and evaluation metrics

We test GPT-3.5 (gpt-3.5-turbo-0301), GPT-4 (gpt-4-0314), Llama2-7B, Llama2-13B, Llama2-70B, and CodeLlama-34B. The decoding parameters we use are: frequency penalty = 0.0, presence penalty = 0.0, temperature = 1.0, top p = 1.0. All Llama models are Llama-Chat models and the CodeLlama model is the Instruct variant. The context window sizes for these models are 4,096 tokens except for GPT-4, which has 8,192 tokens. We focus on zero-shot experiments, where we used the following system prompt: "You are given a task to solve. Make sure to output an answer after "Answer:" without any explanation."

To ensure consistent evaluation, we utilize the following protocol: We first check if the generated text contains the keyword "Answer:". If present, we consider the subsequent text as the model's prediction. In situations where there are multiple ground-truth answers, we store the answers as a set using "," as the separator. We consider the prediction correct only when the generated set of answers matched the ground-truth set exactly.

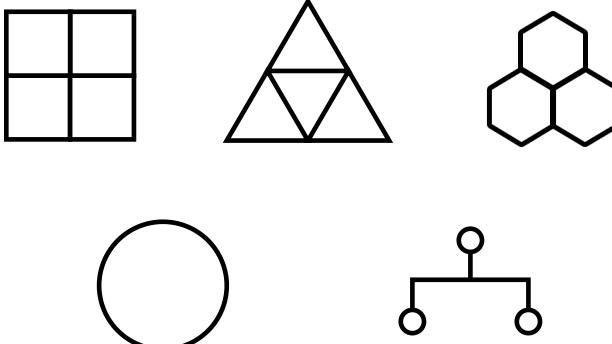

Figure 1: The spatial structures we examine for the underlying maps include squares, triangles, hexagons and rings. Additionally, we analyze a tree structure to explore its relational nature.

**Question**: "You have been given a 2 by 2 square grid. Starting from a vertex, you will move along the edges of the grid. Initially, you are positioned at the bottom left corner of the grid, where you find a box turtle. You move right by one step, where you find a table lamp. You move up by one step, where you find an American black bear. You move left by one step, where you find a hand plane. You move down by one step. What will you find?"
**Answer**: "box turtle"

(a) Square

**Question**: "You have been given an equilateral triangular tile map consisting of 2 rows, where the first row has one tile and the second row has three tiles. Starting from a vertex, you will move along the edges of these tiles. Initially, you are positioned at the bottom left corner of the map, where you find a box turtle. You move right by one step, where you find a hand plane. You move up-right by one step, where you find a guacamole. You move down-right by one step, where you find a table lamp. You move left by one step. What will you find?"
**Answer**: "hand plane"

(b) Triangle

**Question**: "You have been given a pointy-topped regular hexagonal tile map consisting of 1 tile. Starting from a vertex, you will move along the edges of the tile. Initially, you are positioned at the top corner of the map, where you find an ice pop. You move down-right by one step, where you find a Boxer. You move down by one step, where you find a poke bonnet. You move down-left by one step, where you find a combination lock. You move up-left by one step, where you find a spotlight. You move up by one step, where you find a gibbon. You move up-right by one step. What will you find?"
**Answer**: "ice pop"

(c) Hexagon

**Question**: "You have been given a circular grid consisting of 4 connected dots. Starting from a vertex, you will move along the edges of the circular grid. Initially, you are positioned on the dot that's located at the top of the grid, where you find a palace. You move around the ring by 1 step in a clockwise direction, where you find a gong. You move around the ring by 2 steps in a clockwise direction, where you find a shopping basket. You move around the ring by 2 steps in a clockwise direction. What will you find?"
**Answer**: "gong"

(d) Ring

Figure 2: Example question and its answer for square, triangle, hexagon and ring structure.

# 3 Results

## 3.1 Do different spatial structure features affect model performance?

In Section 2, we provide an example that utilizes a square grid to assess the understanding of spatial structures in LLMs. In the example, we exploit the concept of loop closure within the square grid for this purpose. Since loop closure also exists in other spatial structures, we also included triangles, hexagons and rings to explore the spatial understanding ability of LLMs on less common 2D structures. Example prompt is in Figure 2. In this section, we are interested in analyzing how different factors contributes to the "difficulty" of a problem instance (via its prediction accuracy). There are two sources of difficulties: extrinsic difficulty, like the number of navigation steps [2] in the prompts, which is independent of the specific structures; and intrinsic difficulty, which includes high-level structure features, like the structure type, and low-level structure features, like the number of edges and vertices.

With these goals, we run a logistic regression analysis to examine what factors influence the prediction accuracy. Specifically, using the square type as the reference level for graph types, we analyze the prediction outcomes via the following model:

Prediction correctness $\sim$ intercept $+ \mathbf{1}$(graph type == hexagon) $+ \mathbf{1}$(graph type == triangle) $+ \mathbf{1}$(graph type == ring) $+$ number of edges $+$ number of navigation steps.

Note that we only pick the number of edges as our low-level graph feature because the correlation between the number of edges and the number of vertices is very high (correlation coefficient was 0.998). We chose logistic regression as our model because the Prediction correctness is a binary variable (for every prompt, the prediction is either correct (Prediction correctness $= 1$) or wrong (Prediction correctness $= 0$)). We collect a total of 6,100 prediction results of different prompts using GPT-4 varying the structure type, the number of edges, and the number of navigation steps (i.e. hexagon: 1400 samples, ring: 1500 samples, square: 1800 samples, and triangle: 1400 samples). The summarized results are presented in Table 1.

|  | Estimate | Std. Error | z value | Pr($>$\|z\|) |
|---|---|---|---|---|
| (Intercept) | 3.448 | 0.142 | 24.273 | $<$2e-16 (***) |
| type is hexagon | -2.327 | 0.091 | -25.595 | $<$2e-16 (***) |
| type is triangle | -1.820 | 0.082 | -22.199 | $<$2e-16 (***) |
| type is ring | -2.117 | 0.103 | -20.616 | $<$2e-16 (***) |
| number of edges | -0.002 | 0.002 | -1.062 | 0.288 |
| number of navigation steps | -0.345 | 0.018 | -18.924 | $<$2e-16 (***) |

Table 1: Logistic regression results. We see that the number of edges (an example of lower-level features) is not a significant predictor variable for prediction correctness (p value $= 0.288$). However, the higher-level graph structure (e.g. square or hexagon) is a significant predictor of correctness (all p values $<$ 2e-16).

We observe that the prediction accuracy is not significantly influenced by lower-level structure features (the number of edges). However, it does depend on the structure type. Furthermore, the accuracy is heavily influenced by the number of navigation steps (p value $<$ 2e-16). This finding aligns with our intuition, as longer navigation poses greater challenges in tracking objects in a spatial map.

In the above, we model the difficulty of navigation tasks as a function of both extrinsic and intrinsic difficulty attributes. Now we evaluate the difficulty of the task as a function only of the graph structure, separately from the extrinsic difficulty. To achieve this, we condition on the extrinsic difficulty by fixing the number of navigation steps (to be 8) in order to get a measure of the "intrinsic difficulty" of each structure. We use a 3 by 3 square grid, size 2 hexagonal grid, size 3 triangular grid, and size 12 ring grid as the underlying maps. The prompts are generated in a similar manner as Figure 2. We test on several LLMs. The results is in Figure 3.

---

[2]Navigation step refers to the movements asked to be performed in the prompt. In the example prompts in Figure 2, the number of navigation step is 3 for square, 4 for triangle, 6 for hexagon, and 3 for ring.

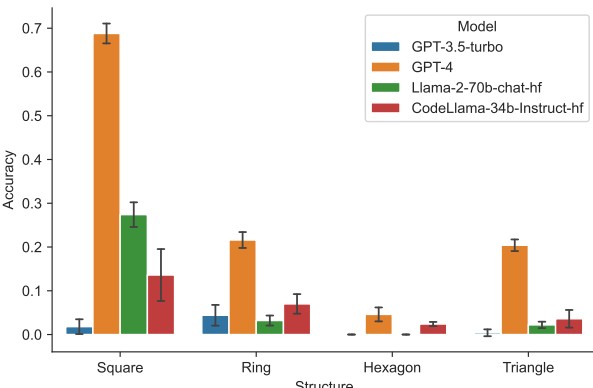

Figure 3: We compare the accuracy of the models across the different spatial structures. The random guessing accuracy is 1/8 since the predictions from random guessing are uniformly selected from the nodes encountered by the models, which corresponds to the local path with 8 navigation steps. GPT-4 have higher prediction accuracy than random guessing in square, ring and triangle structures, but worse in hexagon. ChatGPT exhibits lower prediction accuracy than random guessing across all of these structures. Llama2-70B and CodeLlama-34B shows a similar pattern to GPT-4. The error bars indicate 1.96 times standard error across 5 runs.

When comparing GPT-3.5-turbo and GPT-4, we find that GPT-3.5-turbo performs poorly across all spatial structures, while GPT-4 shows higher variation in performance. GPT-4 excels on the Square structure, ranks second best on the Ring and Triangle structures, and performs the worst on the Hexagon structure. Llama2-70B and CodeLlama-34B, while generally performing worse than GPT-4, exhibit a similar pattern of performance to GPT-4. We omit Llama2-7B and 13B from our discussion because they achieve zero (or very close to zero) accuracy across all structures. This indicates that tackling zero-shot spatial reasoning tasks may necessitate larger models.

The relative ease of the square grid in comparison to other grid types could be attributed to factors such as the prevalence of tabular data and city grid navigation within the model's training data. In addition, coding problems related to maze exploration often involve navigating a two-dimensional square grid, while triangular and hexagonal grids are less commonly encountered. Thus, it is conceivable that such exposure during pre-training makes GPT-4 possess an enhanced understanding of 2D square grids. Analogously, for humans, individuals who grow up in cities with a more grid-like structure may exhibit greater difficulty in navigating through less organized environments, such as older European cities, and vice versa (Coutrot et al., 2022). Additionally, we perform an experiment using the rhombus grid, which was achieved by rotating the square grid 45 degrees. Under the same experimental condition, we find that GPT-4 maintains an accuracy of 0.66, which is slightly lower than the original square-grid accuracy of 0.71 but still significantly higher than other structures. This outcome provides further confirmation that the specific grid structure with two axes contributes to its strong performance.

## 3.2 Is building a local map more difficult than building a full map and retrieving a path?

In the previous section, LLMs were tasked with constructing a local map gradually as they received new information, one step at a time, which we refer to as the "local" setting. Alternatively, we can provide LLMs with the complete map from the start and instruct them to begin exploration from a randomly selected initial location for a specific number of steps, which we refer to as the "global" setting. On one hand, local exploration may be deemed easier as it requires retaining less information. On the other hand, presenting the global map upfront could potentially aid in more accurate map navigation. To address this question, we compare the local and global settings in our spatial understanding task. For this comparison, we use square and ring structures since there are widely accepted methods of specifying global coordinates, making it easier to specify paths from a randomly selected initial position. In particular, we provide the global map

information in the following manner: For the square structure, we list the object names row by row. As for the ring structure, we list the object names starting from the top and proceed clockwise. We then have the model follow a fixed number of navigation instructions, just like the local setting. Example prompts for the square and ring structure under the global setting are in Figure 4.

**Question**: "You have been given a 3 by 3 square grid. In the 1st row, from left to right, we have tiger, rifle, and English Setter. In the 2nd row, from left to right, we have soup bowl, mongoose, and ski. In the 3rd row, from left to right, we have stopwatch, bow, and wolf spider. You start at soup bowl, then you go down by one step, then you go right by one step, then you go right by one step, and then you go up by one step. What will you find?"
**Answer**: "ski."

**Question**: "You have been given a circular path consisting of 12 connected dots. At the start, you are positioned on the dot that is located at the top of the path, where you find a mop. Moving in a clockwise direction from the mop, the elements on the path are a home theater, a limousine, a vacuum cleaner, a Bedlington Terrier, a fireboat, a red panda, a projector, a seashore, a Redbone Coonhound, a Scottish Terrier, and a laptop computer. Starting from the laptop computer, you move around the ring by 12 steps in a clockwise direction. What will you find?"
**Answer**: "laptop computer."

(a) Square

(b) Ring

Figure 4: Example question and its answer for square and ring structure under the global setting.

The results in Figure 5 show that the global setting is slightly harder than the local setting for both 3 by 3 square and size-12 ring structures (all under 8 navigation steps), except when the performance is already low for the ring structure for Llama2 models, which shows a less clear pattern as such.

The results presented in Figure 5 show that, in general, the global setting is more challenging than the local setting for both the square and the ring structures. However, this trend becomes less evident when considering the already low performance of Llama2 models on the ring structure, leading to a less clear pattern in this case.

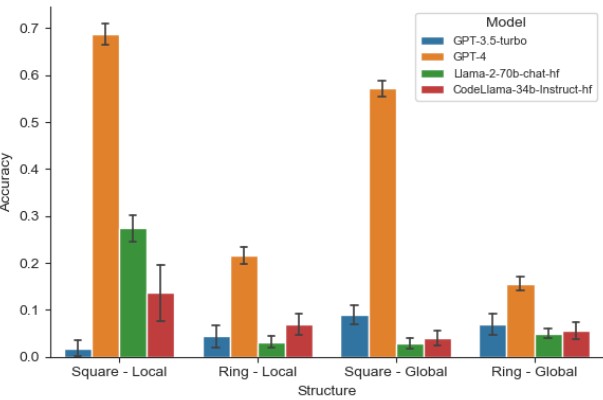

Figure 5: Performance is evaluated on GPT-4, Llama2-70B, and CodeLlama-34B. For both square and ring structures, we observe that the prediction accuracy of GPT-4 using the local map is higher compared to the global map. Llama2-70B and CodeLlama-34B show a similar pattern for the square, while the pattern is less clear for the ring. The error bars indicate 1.96 times standard error across 5 runs.

### 3.3 The order of presenting the map impacts spatial understanding

In the previous section, our approach to providing complete map information upfront to the model has involved a specific method. We describe the items in the map row by row, indicating their positions from left to right. For example, in the first row, we have item A, item B, and item C. In the second row, we

have item D, item E, and item F from left to right, and so on. However, there exist multiple ways to convey the same information. Here, we explore different approaches to feeding data into the model. In addition to the aforementioned method, we examine two alternative techniques: random and snake order. The random approach involves placing items in the map at random positions using the global coordinate system. On the other hand, the snake order method follows a specific pattern. In the first row, items are fed from left to right as before. However, when transitioning to the second row, we introduce the instruction "you move down by one step" to indicate the change in row. In the second row, items are then fed from right to left. By investigating these alternative data feeding methods, we aim to understand their implications and assess their impact on the model's performance. Example prompts using random and snake order are shown in Figure 6.

**Question**: "You have been given a 3 by 3 square grid with various items located at different indices: a restaurant is at index (1, 2), a half-track is at index (2, 2), a sleeping bag is at index (2, 3), a Scottish Terrier is at index (2, 1), a wok is at index (1, 3), a balance beam is at index (3, 3), a military uniform is at index (3, 1), a marimba is at index (1, 1), and a radio telescope is at index (3, 2). You start at the position where the half-track is located, then you go left by one step, then you go up by one step, then you go right by one step, then you go right by one step, then you go down by one step, then you go down by one step, then you go left by one step, and then you go up by one step. What will you find?"
**Answer**: "half-track."

(a) Random order

**Question**: "You have been given a 3 by 3 square grid. Starting from a vertex, you will move along the edges of the grid. Initially, you are positioned at the bottom-left corner of the grid, where you will find a toilet paper, then you go right, where you will find a stethoscope, then you go right, where you will find a screw. Then you go up, where you will find a modem, then you go left, where you will find a pretzel, then you go left, where you will find an Otterhound. Then you go up, where you will find an Asian elephant, then you go right, where you will find a tile roof, then you go right, where you will find a slip-on shoe. Now you have all the information on the map. You start at the position where the pretzel is located, then you go right by one step, then you go down by one step, and then you go up by one step. What will you find?"
**Answer**: "modem."

(b) Snake order

Figure 6: Example question and its answer for 3 by 3 square grid using random and snake order.

The results are shown in Table 2. Although the GPT-4's accuracy degrades for Random and Snake, it is noteworthy that Random is better than Snake. We note that we omit the results for Llama-2 models because even Llama2-70B's performance was already very low (e.g. row-by-row's accuracy is 0.04 for Llama2-70B whereas GPT-4 achieves 0.55.)

|  | Row-by-row | Random | Snake | Snake+Coord |
|---|---|---|---|---|
| GPT-4 Acc | 0.572 (0.019) | 0.495 (0.007) | 0.440 (0.057) | 0.525 (0.049) |

Table 2: The order of presenting the map impacts spatial understanding accuracy. Snake+Coord refers to the setting where we append the global coordinates of the location after each step. The mean and the standard deviation (shown in parentheses) are calculated over 5 different runs.

What could potentially explain these phenomena? It is reasonable to speculate that different methods of inputting data could influence how LLMs internally represent spatial relations. For instance, when utilizing the row-by-row approach, the LLM can register these items in a key-value dictionary, where the row id serves as the key and a list of objects represents the corresponding value. The 'random' approach also enables the LLM to store a key-value dictionary where the key denotes the location address and the value denotes a single item, which can be leveraged for navigation purposes later on. On the other hand, the 'snake' order approach necessitates the LLM to simultaneously handle the storage of object item information and spatial relational understanding. This added complexity potentially complicates the task.

To investigate whether or not such a key-value data structure plays a role, we perform an additional experiment where we add the global coordinate information to the snake order approach. We see that this approach indeed increases the accuracy from 0.40 to 0.56, corroborating our hypothesis.

## 3.4   Relational structure: Tree

In addition to spatial structures, relational structures can also be represented using connected graphs. In this section, we examine a tree structure. Unlike spatial structures, the hierarchical structure of a tree is

more naturally presented in a global setting, where all objects are provided at the beginning, and relational questions can be asked subsequently.

To ensure comparability with the square and ring structures, we set the number of nodes in each tree to be 9. For comparison, we also included a 3 by 3 square grid and a 9-node ring in this experiment. The exploration steps are set to 4 for all structures. For the tree structure, we utilize the same ImageNet object labels, but focus on relational questions that involve 4 steps, such as "What is the cousin of A?", "What is the great-great-grandparent of A?", and "What is/are the great-great-grandchild/children of A?". An illustrative example question and its corresponding answer can be found in Figure 7.

**Question**: "You have been given a tree structure with 9 nodes. The root node is a great white shark. The great white shark has 2 children: a garter snake and a Gila monster. The garter snake has 2 children: a jigsaw puzzle and a moped. The jigsaw puzzle has a child: a Tibetan Terrier. The Tibetan Terrier has no children. The moped has a child: an umbrella. The umbrella has no children. The Gila monster has 2 children: a Christmas stocking and a horse-drawn vehicle. The Christmas stocking has no children. The horse-drawn vehicle has no children. What is the cousin of the moped? "
**Answer**: "Christmas stocking, horse-drawn vehicle"

Figure 7: Example prompt for tree structure.

The evaluation results are depicted in Figure 8. We observe that for GPT-4, while the tree structure performs worse than the square structure, it outperforms the ring structure. On the other hand, for GPT-3.5-turbo, the tree structure exhibits better performance compared to the square structure. We observe that just like GPT-3.5-turbo, Tree performs better than Square for all Llama-2 models. The only exception is GPT-4; this further demonstrates that GPT-4's exception ability to comprehend the square structure. We also note that Ring is harder than Square for all Llama models and GPT-4. Further investigation into how relational structure, spatial structure, and model size impact performance would be an intriguing topic for future research.

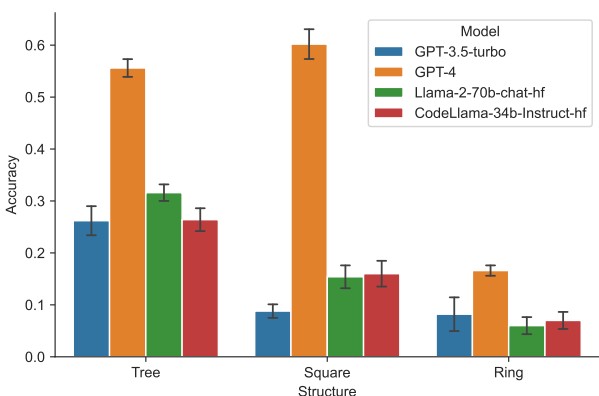

Figure 8: We evaluate the prediction accuracy of the models on a 9-node tree, a 3 by 3 square, and a 9-node ring structure with 4 exploration steps in the global setting. Comparing the performance of GPT-4 and random guessing, GPT-4 outperforms random guessing with higher prediction accuracy, with the order of accuracy being square > tree > ring. GPT-3.5-turbo also performs better than random guessing on the tree structure, but worse on the square and ring structures, with the order of accuracy being tree > ring > square. Just like GPT-3.5-turbo, Tree performs better than Square for all Llama-2 models. The error bars indicate 1.96 times standard error across 5 runs.

### 3.5 Grid size inference from sequences of navigational instructions

In the preceding sections, we examined the capability of LLMs to understand the spatial and relational structure of a map. In this section, our focus shifts to investigating whether LLMs can infer the global size of a map based solely on a sequence of local navigational actions. Simultaneously inferring the extent of

map while navigating is an important component of navigation as a cognitive task. Inferring the extent of a map is a non-trivial task which requires integrating information about the history of actions and reasoning about what it implies spatially. Indeed, humans often have a sense of the size of the places they visit, not only how to navigate from point A to point B. Specifically, we provide navigational instructions that guide the exploration of all locations within a rectangle. As before, we also provide what item the agent finds at each step. Then we ask LLMs about the height and width of the rectangle. This task necessitates LLMs to maintain the entire path in order to accurately deduce the overall dimensions of the rectangle.

Table 3 illustrates the accuracy comparison of GPT-4 for different size configurations of the same area (e.g., 2 by 6, 3 by 4 for an area of 12). We prepare 200 samples for each area. We observe a general trend where accuracy decreases as the length of the sides increases and as the area size of the rectangular grid increases. We omit the results for GPT-3.5-turbo, Llama-2-70B and CodeLlama-34B because these models were not able to infer the size of rectangle.

|  | 3x4 or 4x3 | 2x6 or 6x2 | 4x6 or 6x4 | 3x8 or 8x3 | 2x12 or 12x2 |
|---|---|---|---|---|---|
| GPT-4 Acc | 0.612 (0.209) | 0.103 (0.058) | 0.162 (0.071) | 0.016 (0.017) | 0.005 (0.007) |

Table 3: Grid size inference performance of GPT-4. The mean and the standard deviation of prediction accuracy (shown in parentheses) are calculated over 10 different runs.

Additionally, we evaluate the same setup but exclude object item information during navigation. In this case, GPT-4 relies solely on directional information (e.g., "you go up by one step, then you go down by one step, etc."). Shown in Figure 9 are the mean accuracy and standard deviation across 10 different runs, grouped by the shape of the rectangles. We see variations in accuracy across based on the shape of the rectangles.

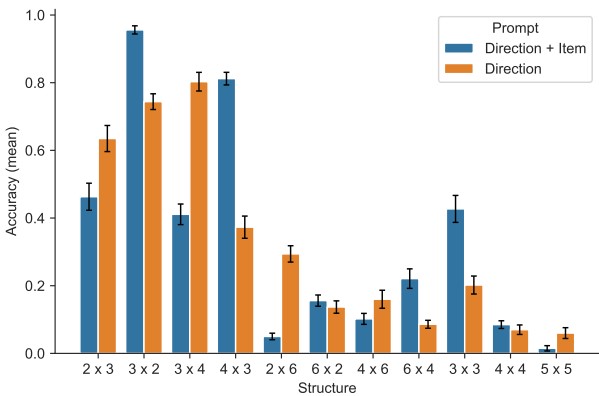

Figure 9: For grid size inference, we see variations in accuracy based on the shape of the rectangles, specifically whether they were 'tall' or 'fat'. (e.g. "4x3" is consistently better than "3x4" for prompts that combines both directional and object item information at each step, while "3x4" is consistently better than "4x3" for the approach that only uses directional information.) The error bars indicate 1.96 times standard error across all runs.

## 4 Error analysis

In this section, we perform a detailed analysis of the errors produced by GPT-4, to assess whether it is modelling the correct topology. In our error analysis, we focus on GPT-4 because of its relatively strong performance, which reveals intriguing error patterns.

To study the extent to which LLM understands the topology of a given spatial structure, we examine what types of mistakes it makes. In our spatial understanding task, when LLM makes a mistake, in the overwhelming majority of cases, it provides the name of an object at a different location (rather than naming an object that did not appear at all in the prompt). Therefore, we can measure the distance between the

correct location and the location of the predicted object with respect to the underlying topology. If the LLM represents the spatial structure of the map, the distribution of these distances will tend to cluster at small values, which we call spatial-topology bias. That is, if the LLM represents spatial structure, we should expect more mistakes for objects topologically close to the correct location, and fewer mistakes for objects farther away. A natural choice for measuring distances in grids is the shortest distance between two vertices in the grid.

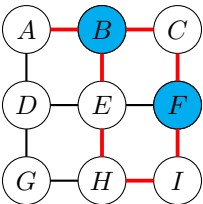

Figure 10: An example of two distance metrics in $3 \times 3$ square grid in the local setting. If the path in the local setting starts at node A and follows through B, C, F, I, H, E, and finally ends at B, and if the LLM predicts F instead of B, then the temporal distance between B and F is 4, while the spatial distance is 2.

LLMs may also show non-topological biases in their predictions, an instance of which is the temporal bias – an inclination to predict objects that are observed in the textual vicinity of the ground truth item in the given prompt. To investigate the presence of this bias, we measure the temporal distance as the number of objects between the first occurrence of the ground truth in the prompt and the predicted item (including the predicted item). An illustrative example of the temporal and spatial distances in the local setting is shown in Figure 10.

In the following, we examine the error distributions of GPT-4 in square, triangular, and hexagonal grids. For each experiment, we collect 1000 predictions from GPT-4, and analyze them along with their corresponding prompts and ground-truth correct answers. Out of the 1000 prompts, we only consider the subset of predictions in which the model was wrong. We compare GPT-4's errors with the random distributions of spatial and temporal distances with respect to a uniform baseline. To do this, we randomly pick one of the prompts the model is tested with. Next, we record the location of the ground truth object based on the prompt. Then, for the global setting prompts, we at random select a location from the entire grid; for the local setting prompts, we at random select one of the visited nodes along the path. We then calculate both the spatial or temporal distances between this randomly selected location and the ground truth location. We repeat this procedure for 100,000 times to generate the error distribution for the uniform baselines.

## 4.1 Comparing error distributions of square, triangular, and hexagonal grids

In our analysis, we used a 3 by 3 square grid, a triangular grid with a size of 3, and a hexagonal grid with a size of 2. We chose these grid configurations to ensure a fair comparison across different grid structures, allowing for prompts conducting 8 navigation steps in each grid. The precise shape of the size-3 triangular grid is shown in Figure 15 in Appendix, and the shape of size-2 hexagonal grid is shown in Figure 1. The results for the local setting are shown in Figure 11.

For the square grid, GPT-4 tends to make more errors at spatial distances of 1 relative to random baseline, indicating a spatial-topology bias (Fig. 11a, left). However, temporal distance also shows a stronger peak at the value of 1 (relative to the uniform baseline; Fig. 11a, right) indicating that having two items more closely located in the prompt is also an effective predictor of GPT-4's errors. In Appendix Figure 19, we also generated a plot for the conditional distribution of TD when SD=1 to further validate the temporal bias.

In the case of the hexagonal grid (Fig. 11b), we see a lack of spatial-topology bias, with the distribution of distances peaking at 2 (instead of 1). We also do not see a temporal bias in GPT-4's behavior, again with a distribution peaked at the temporal distance of 2. This indicates the presence of some other source of non-spatial bias besides the temporal bias. Closer inspection revealed that whenever GPT-4 makes an error in hexagonal grids, these errors are often due to the model predicting the very first object on the path, which often ends up having spatial and temporal distances of 2 from the ground truth correct answer.

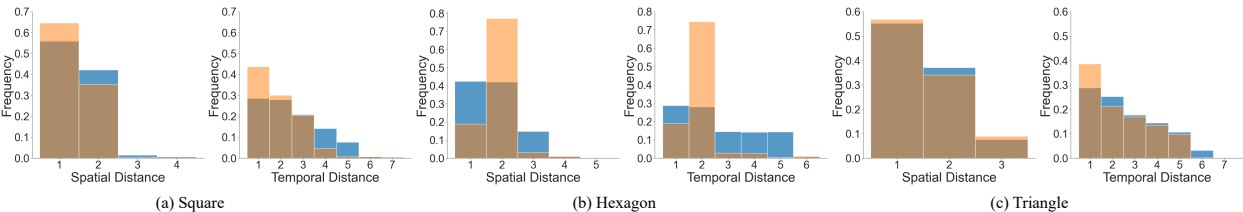

Figure 11: The spatial and temporal distance (SD, TD, respectively) histograms for square, hexagonal, and triangular grids under the local setting. Blue histograms show random baselines. Orange histograms show the observed distribution of errors as the spatial (left) and temporal (right) distances between the ground truth and GPT-4's predicted locations. (a) In grids with square topology, GPT-4 makes more errors both when SD is 1 and TD is 1, compared to the uniform baseline, meaning that both spatial and temporal biases contribute to GPT-4's errors. (b) In grids with hexagonal topology, we do not observe spatial nor temporal bias. (c) The simultaneous lack of spatial bias and the presence of temporal bias indicate that GPT-4 was not able to accurately construct the triangular grid.

For the triangular grid (Fig. 11c), the distribution of SD from GPT-4 and random guessing is almost the same, suggesting that there is almost no spatial bias. However, there is a spike when TD = 1. We find that among all the instances where TD equals 1, the proportion of predicting the starting position is 0.416. Hence, it appears that the temporal bias accounts for more than half of the bias observed in the triangular grid. In terms of the starting position bias of the square grid, see Appendix Section C.

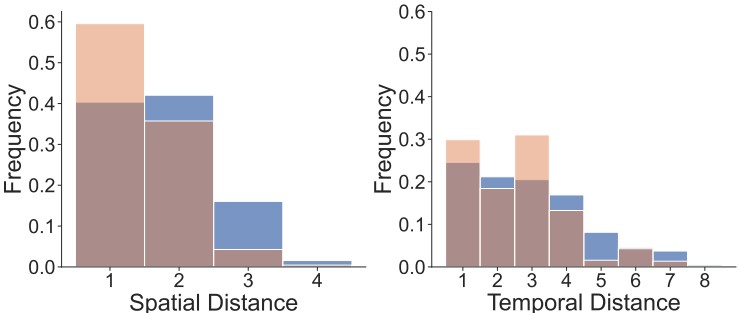

Figure 12: The spatial and temporal error histograms for the 3 by 3 square grid under the global setting. We see a spike when the temporal distance (TD) is 3 and 6, indicating an effect of spatial distance – these two TD values correspond to the spatial distance of 1 and 2.

In Figure 12, we present analyses for the global structure.[3] When dealing with a square grid, we provide GPT-4 with the complete map upfront by listing each object row by row. For instance, in prompts for the square grid structure in the global setting, such as "In the first row, we have item A, B, and C. In the second row, we have item D, E, and F, ..." the temporal distance between A and D would be 3. If GPT-4 represents the square grid as a one-dimensional array, we expect that the frequency of temporal distance would decrease steadily. However, we observe spikes at 3 and 6 in the temporal distance, which correspond to spatial distances of 1 and 2, respectively. This finding suggests that GPT-4 makes more errors when objects are closer to the ground truth in terms of spatial structure, rather than temporal distance. It supports the notion that GPT-4 actually models some aspect of the two-dimensional structure. Furthermore, we investigate whether there are any discrepancies in error distributions when the distance is calculated using either row-major order or column-major order. Figure 13 demonstrates that there is an almost symmetrical distribution between rows and columns. This provides additional evidence that GPT-4 recovers the structure of the two-dimensional array.

---

[3]We omit the ring structure because each movement can involve many steps around the ring (e.g. "move by 3 steps clockwise"), which effectively introduces edges between all graph nodes, and thus renders analysis based on spatial distance less meaningful.

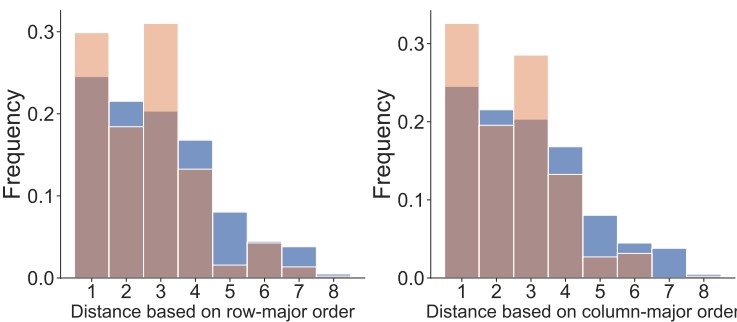

Figure 13: The row-wise and column-wise error histograms for the 3 by 3 square grid with 8 navigation steps under the global setting. We see that both row and column-wise histograms are almost identical, which suggests GPT-4 does not have bias for row or column.

## 5 Comparison to human baseline

Finally, we have conducted human experiments to assess the average human baseline performance. These experiments focused on local navigation tasks using the structures shown in Figure 3, which include a 3 by 3 square grid, a size 2 hexagonal grid, a size 3 triangular grid, and a size 12 ring grid. For each of these structures, we randomly selected 20 prompts from the previously used dataset, resulting in a total of 80 candidate prompts. Then, for each participant in the experiment, we randomly chose 10 prompts from this pool of candidate prompts. Participants were asked to provide textual answers to these questions. Additionally, we included 4 attention check questions. These questions were intentionally designed to be easy so that we can assess whether participants were providing meaningful answers, and were used as an exclusion criterion in our analysis. These attention check questions were drawn from the set of questions associated with very simple structures, such as a 2 by 2 square grid, a size 1 hexagonal grid, a size 2 triangular grid, and a size 5 ring grid. These attention check questions were distributed randomly throughout the survey.

We established a criterion to measure participants' engagement in the experiment: if a participant made more than one mistake in the attention check problems, we would exclude their response from the analysis. Specifically, each participant had 30 minutes to solve 14 problems, which consisted of 10 regular problems to evaluate human performance and 4 attention check problems. The participants were not informed whether the given problem was a regular task or an attention check.

In this experiment, we received completed responses from a total of 23 participants. Details about the participant recruitment process and the selection criteria are discussed in Appendix Section F. Among them, 5 participants did not meet the attention check criterion mentioned earlier. Their responses were excluded, leaving us with the responses of the remaining 18 participants for analysis. In total, we collected 180 question-answer pairs, distributed as follows: 48 pairs for the 3 by 3 square grid, 41 for the size 12 ring grid, 48 for the size 2 hexagon grid, and 43 for the size 3 triangle grid. We converted all the responses to lowercase and removed all English articles (a, an, the) from both the human responses and the ground truth. A response was considered "correct" only if it matched the ground truth exactly. The results can be found in Table 4. Similar analysis under a different criterion of participants' engagement measurement is conducted with results in Section F.

|  | Square | Ring | Hexagon | Triangle | Aggregated |
|---|---|---|---|---|---|
| Human | 0.90 | 0.78 | 0.41 | 0.58 | 0.67 |
| GPT-4 | 0.69 | 0.22 | 0.05 | 0.20 | 0.29 |

Table 4: Human baseline performance compared against GPT-4. The values of the GPT-4 accuracy are taken from Figure 3.

The aggregated accuracy across all the structures is 0.67, and the accuracy for each structure is 0.90 for 3 by 3 square grid, 0.78 for size 12 ring grid, 0.41 for size 2 hexagon and 0.58 for size 3 triangle grid respectively. For GPT-4, the accuracies for each structure are from Figure 3. Although human responses

are not perfect, they still outperform GPT-4 by a significant margin. It is also interesting to note that, like GPT-4, non-expert humans struggle with non-square grid shapes. The differences in human performance across different structures may be attributable to familiarity with such structures in every-day life and how well these structures can be described in natural language. For example, in contrast to the square and ring structures, hexagonal and triangular grids are much less frequently encountered in everyday life. Moreover, these structures are intrinsically more difficult to describe in natural language.

## 6  Related Work

Some research suggests that language models have the ability to acquire implicit world models (Abdou et al., 2021; Patel & Pavlick, 2022; Li et al., 2021). Spatial understanding is particularly intriguing because it might seem counterintuitive that a language model, which lacks visual or sensorimotor input, can comprehend spatial structures.

Patel & Pavlick (2022) provide evidence that GPT-3 is capable of grounding spatial and cardinal direction terms in a text-based grid world. They present contextual examples of cardinal directions (e.g., north, east, northeast) and evaluate whether the model can generalize to a different subset (e.g., south, west, southwest). Our work expands upon these findings by assessing spatial understanding that necessitates the accurate construction and retention of representations of spatial structure in more challenging tasks.

Previous studies have employed text-based navigation tasks to evaluate language models. Bubeck et al. (2023) evaluate GPT-4 across various domains, including mathematics, coding, vision, medicine, law, and psychology. In one task involving embodied interaction, they create a simple map and prompt GPT-4 to explore it interactively using actions such as left, right, up, and down. A human provides feedback during the exploration. They demonstrate that GPT-4 successfully tracks all the locations and visualizes them using a generated program. However, their study only examines a single instance of a square-grid map and does not thoroughly investigate the extent of GPT-4's spatial understanding. Another similar task is present in (Whittington et al., 2020) but the goal is to test structural generalization. BIG-bench collaboration (2023) has a task where the agent is required to determine whether it would return to its original starting position based on a set of navigation instructions. However, this task only involves providing "yes/no" answers. In the NLP community, more complex spatial reasoning tasks have been introduced, including those involving multi-hop reasoning (Shi et al., 2022) and tasks that involve understanding of textual descriptions for intricate visual scenes (Mirzaee et al., 2021). In contrast, our study investigates various spatial structures, such as rings, trees, hexagons, and triangles, while also requiring the language model to remember and track object names.

Moreover, LLMs increasingly underlie embodied agents, and exhibit impressive spatial reasoning capability (Singh et al., 2023; Liang et al., 2023; Huang et al., 2022; Rajvanshi et al., 2023). Our work explores the basic spatial reasoning skills of these models via controlled experiments using synthetic grid data. This approach allows us to identify both their strengths and weaknesses.

A concurrent work (Momennejad et al., 2023) also evaluates LLMs in terms of cognitive mapping and planning abilities. While they examine a wide array of tasks, their focus is on planning (e.g. generating path between given vertices), and their graph structures are limited to trees, linear paths, and social graphs. In contrast, our work is centered on path integration for sensory prediction and perform a systematic evaluation of basic topological spatial structures, coupled with in-depth error analysis with experiments on human participants.

## 7  Conclusion

In this work, we investigated whether LLMs could build representations of spatial structure implicitly from their sequential inputs. We found that LLMs are able to answer questions about spatial relationships, and tend to make errors that reflect spatial proximity. However, the details of their performance depends on the task and structure — square grids are easier than other structures, and local presentation is easier than global. Overall, our results suggest that LLMs implicitly learn to represent aspects of spatial structure,

though their performance is far from perfect. These findings contribute to the growing literature on the aspects of world knowledge that LLMs implicitly acquire from their language-only training.

## Limitations

In our investigation into the spatial understanding of LLM, our focus lies on the zero-shot setting. We have conducted a preliminary study to examine the impact of chain-of-thought style prompting, and we observe an increase in performance for GPT-3.5. However, beyond the 3-shot setting, the performance improvement reached a plateau. Details of these studies can be found in Fig 14 in Appendix, and we encourage future research to explore the effects of additional variations of chain-of-thought prompting on LLM's spatial understanding. We note that the lack of complete detail about the training of GPT-4 makes understanding the origins of its strong spatial performance somewhat challenging. Investigating how smaller models fine-tuned on spatial tasks exhibit spatial understanding would be an intriguing subject for further study.

## Broader Impact Statement

Before the human experiment starts, all participants are provided with an online form with information about the experiment, including the purpose, a brief description, the procedure, confidentiality, and voluntary participation. The participants are informed that their responses will be confidential and they are free to decline or end participation at any time. In the "agreement to participate" section, we need participants to confirm that by clicking the "start" button, they acknowledge that they have read the above information, and agree to participate in the study. The experiment only starts after they click the "start" button.

The participants were recruited from an online crowdsourcing platform (Prolific.com), it is a platform for researchers to post behavioral studies and recruit participants. We set the only participants' criteria to be "fluent in English" as all our prompts and questions are written in English. Our study is evenly distributed to all available participants. The demographic data of the 23 participants that took part in our study is as follows: age range (min: 21, max: 62, median: 36, mean: 41.9), gender (Male: 12, Female: 10, No data: 1), employment status (Full-time: 9, Not-in-paid-work: 5, part-time: 3, unemployed: 2, No data: 4). Although we seek to mitigate the bias by setting as few criteria as possible, there is still a bias toward English-speaker with computers and internet connection, as well as some characteristics of the WEIRD (Western, Educated, Industrial, Rich, Democracies) population.

As large language models continue to advance and find applications in real-world scenarios, it becomes increasingly crucial to assess the risks and unintended consequences associated with such LLM-based applications. Although our study on the spatial understanding of LLMs may not directly address the reduction of harm and bias in these models, we believe it is important to comprehend their inner workings. We hope that our work contributes to the ongoing effort of understanding and exploring the mechanisms at play within LLMs.

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

## A    Additional prompt examples

We provide additional examples of prompts we used with bigger size for different structures in Figure 14.

## B    Temporal distance in the global setting

For instance, in prompts for the 3 by 3 square grid structure in the global setting, such as "In the first row, we have item A, B, and C. In the second row, we have item D, E, and F, ..." the temporal distance between A and D would be 3, while the spatial distance between A and D is 1.

## C    Starting position bias of the square grid

In the 3 by 3 square grid, the ground truth is only present at the 1st, 3rd, 5th, and 7th nodes of the path. To qualify as a starting position bias when Temporal Distance (TD) equals 1, the ground truth would need to occur at the 2nd node of the path. Consequently, there is no contribution from the starting position bias when TD = 1.

## D    Chain-of-thought prompting

In the case of in-context learning experiments, we used the following system prompt: "You are given a task to solve. Make sure to output a final answer after "Answer:"." We include in-context examples in the user prompt in the following format: "Question:\n [question] \n Explanation:\n [explanation] \n Answer:\n [answer]." An example prompt is given in Figure 16.

**Question**: "You have been given a 3 by 3 square grid. Starting from a vertex, you will move along the edges of the grid. Initially, you are positioned at the top right corner of the grid, where you find a breastplate. You move down by one step, where you find a hummingbird. You move down by one step, where you find an eastern diamondback rattlesnake. You move left by one step, where you find a basketball. You move up by one step, where you find a pulled rickshaw. You move left by one step, where you find a Black and Tan Coonhound. You move up by one step, where you find a breakwater. You move right by one step, where you find a sea cucumber. You move right by one step. What will you find?"
**Answer**: "breastplate"

**Question:** "You have been given a 3 by 3 square grid. Starting from a vertex, you will move along the edges of the grid. Initially, you are positioned at the top left corner of the grid, where you find a cock. You move down by one step, where you find a geyser. You move right by one step, where you find a jellyfish. You move up by one step, where you find an impala. You move right by one step, where you find a box turtle. You move down by one step, where you find an espresso machine. You move down by one step, where you find a bib. You move left by one step, where you find a megalith. You move up by one step. What will you find?"
 **Answer:** "jellyfish"

(a) 4 by 4 square grid with 8 navigation steps

**Question:** "You have been given an equilateral triangular tile map consisting of 3 rows, where the first row has one tile, the second row has three tiles, and so on, so that the i th row has 2*i-1 tiles. Starting from a vertex, you will move along the edges of these tiles. Initially, you are positioned at the top corner of the map, where you find an ear. You move down-left by one step, where you find a sundial. You move down-left by one step, where you find a flagpole. You move down-right by one step, where you find a West Highland White Terrier. You move right by one step, where you find a Sussex Spaniel. You move up-right by one step, where you find a howler monkey. You move left by one step, where you find a Basset Hound. You move up-right by one step, where you find an ice pop. You move up-left by one step. What will you find?"
**Answer:** "ear"

**Question**: "You have been given an equilateral triangular tile map consisting of 3 rows, where the first row has one tile, the second row has three tiles, and so on, so that the i th row has 2*i-1 tiles. Starting from a vertex, you will move along the edges of these tiles. Initially, you are positioned at the bottom right corner of the map, where you find a measuring cup. You move left by one step, where you find a binoculars. You move left by one step, where you find a space heater. You move up-right by one step, where you find a bolo tie. You move right by one step, where you find an African bush elephant. You move up-right by one step, where you find a ladle. You move left by one step, where you find an English Setter. You move down-left by one step. What will you find?"
**Answer:** "bolo tie"

(b) Size-3 triangular grid with 8 navigation steps

**Question:** "You have been given a pointy-topped regular hexagonal tile map consisting of 2 rows, where the first row has one tile and the second row has two tiles. Starting from a vertex, you will move along the edges of these tiles. Initially, you are positioned at the bottom left corner of the map, where you find a drilling rig. You move up by one step, where you find a carousel. You move up-right by one step, where you find a loupe. You move up by one step, where you find a gown. You move up-right by one step, where you find a printer. You move down-right by one step, where you find a black-footed ferret. You move down by one step, where you find a station wagon. You move down-left by one step, where you find a bath towel. You move up-left by one step. What will you find?"
**Answer**: "loupe"

**Question:** "You have been given a pointy-topped regular hexagonal tile map consisting of 2 rows, where the first row has one tile and the second row has two tiles. Starting from a vertex, you will move along the edges of these tiles. Initially, you are positioned at the top corner of the map, where you find a bucket. You move down-left by one step, where you find a gown. You move down by one step, where you find a racket. You move down-right by one step, where you find an amphibious vehicle. You move down by one step, where you find a CD player. You move down-left by one step, where you find a T-shirt. You move up-left by one step, where you find a library. You move up by one step, where you find a moped. You move up-right by one step. What will you find?"
**Answer:** "racket"

(c) Size-2 hexagonal grid with 8 navigation steps

**Question:** "You have been given a circular grid consisting of 12 connected dots. Starting from a vertex, you will move along the edges of the circular grid. Initially, you are positioned on the dot that's located at the top of the grid, where you find a milk can. You move around the ring by 4 steps in a counter-clockwise direction, where you find a mushroom. You move around the ring by 9 steps in a counter-clockwise direction, where you find a spotlight. You move around the ring by 1 step in a counter-clockwise direction, where you find a shopping basket. You move around the ring by 5 steps in a counter-clockwise direction, where you find a safety pin. You move around the ring by 8 steps in a counter-clockwise direction, where you find a poke bonnet. You move around the ring by 10 steps in a clockwise direction, where you find a Standard Schnauzer. You move around the ring by 6 steps in a counter-clockwise direction, where you find a shovel. You move around the ring by 4 steps in a clockwise direction. What will you find?"
**Answer:** "safety pin"

**Question:** "You have been given a circular grid consisting of 12 connected dots. Starting from a vertex, you will move along the edges of the circular grid. Initially, you are positioned on the dot that's located at the top of the grid, where you find a giant panda. You move around the ring by 9 steps in a clockwise direction, where you find a pickup truck. You move around the ring by 6 steps in a counter-clockwise direction, where you find a car wheel. You move around the ring by 5 steps in a clockwise direction, where you find a vulture. You move around the ring by 3 steps in a clockwise direction, where you find a quill. You move around the ring by 5 steps in a counter-clockwise direction, where you find a fountain pen. You move around the ring by 11 steps in a counter-clockwise direction, where you find a snoek. You move around the ring by 6 steps in a counter-clockwise direction, where you find a military uniform. You move around the ring by 7 steps in a clockwise direction. What will you find?"
**Answer:** "vulture"

(d) 12-node ring with 8 navigation steps

Figure 14: Example question and its answer for square, triangle, hexagon and ring structures under local setting.

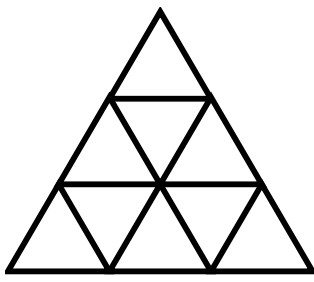

size-3 triangular grid

Figure 15: Size-3 triangular grid.

**Question**: "You are at the top left corner of a 2 by 2 grid, where you find box turtle. You move right by one step, where you find table lamp. You move down by one step, where you find American black bear. You move left by one step, where you find hand plane. You move up by one step. What do you find?"
**CoT**: "We can describe our movements in the 2 by 2 grid starting from the the top left corner as follows:\n- Move right from (1,1) to (2,1)\n- Move down from (2,1) to (2,2)\n- Move left from (2,2) to (1,2)\n- Move up from (1,2) to (1,1)\nAs a result, we reach the coordinate (1,1) where we find the box turtle. Therefore, the answer is box turtle."
**Answer**: "box turtle",

Figure 16: An example of Chain-of-thought style prompting for the 2 by 2 square grid.

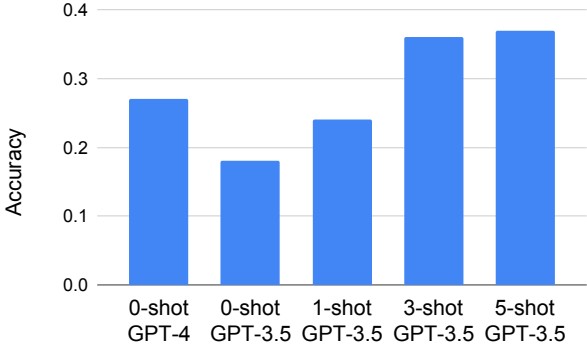

Figure 17: When chain-of-thought style prompting is employed, the accuracy of GPT-3.5 shows improvement as we increase the number of examples provided in the prompt. However, it reaches a plateau when the number of examples ranges between 3 and 5. The setting here is Ring of size 3, the number of navigation steps is 3.

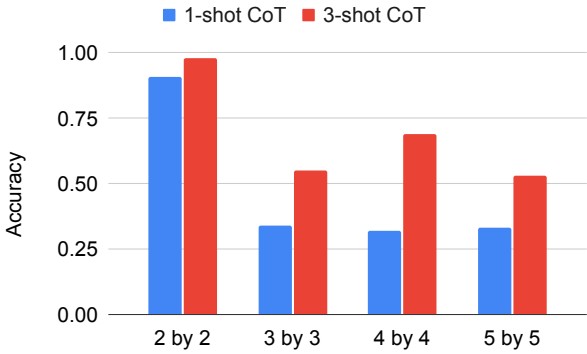

Figure 18: The accuracy for GPT-3.5 vs. the size of square structures in the global setting with navigation steps being 4. Although 3-shot CoT improves over 1-shot CoT, the performance plateaus, as we increase the size of the square structure.

# E   Conditional distribution of temporal Distance

Condition on spatial distance = 1, we can see that the temporal distance has a clear bias towards smaller values in Figure 19 .

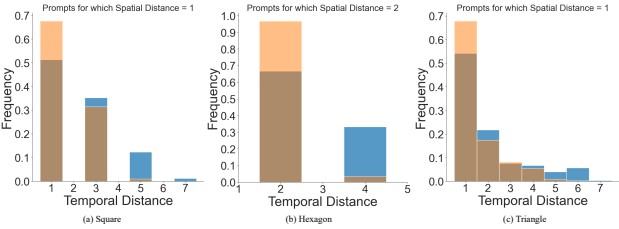

Figure 19: The temporal distance histogram conditioned on spatial distance (SD) = 1 for the square, SD = 2 for the hexagonal, and SD = 1 for the triangular grid under the local setting. These spatial distance conditions are selected because of its maximum frequency in Fig 11.

# F   Human experiments

**Recruitment Process**     We recruited participants online through an online crowdsourcing platform (Prolific.com), it is a platform for researchers to post behavioral studies and recruit participants. The way it works is that we first create a web interface for our study and upload it to Prolific with estimated finish time, hourly payment and participants criteria, then the platform will send out the study invitation to participants who match the criteria. Participants take part in the study online and remotely using their own laptop.

**Impact of the exclusion criteria**     In our experiment, we set the participants' criteria to be fluent in English as answering the prompts required the understanding of English, and the study to be evenly distributed to all available participants. The demographic data of the 23 participants that took part in our study is as follows: age range (min: 21, max: 62, median: 36, mean: 41.9), gender (Male: 12, Female: 10, No data: 1), employment status (Full-time: 9, Not-in-paid-work: 5, part-time:3, unemployed: 2, No data: 4), student status (Yes: 3, No: 17, No data: 3). In the experiment, we seek to mitigate the bias by setting as few criteria as possible, but it is inevitably skewed to people with computers and internet connection, and some characteristics of the WEIRD (Western, Educated, Industrial, Rich, Democracies) population, which is unfortunately still a common practice across much of cognitive science and psychology research.

**Different criterion for measuring participants' engagement**     As one of the reviewer suggests, given the difficulty of non-square grid questions, we conducted similar analysis under the criterion that only

the participants that gets the square attention check wrong will be excluded. Among the 23 participants, only one participant failed the attention check for the rectangular grid, and this participant got 3 out of 4 attention checks wrong. If we exclude this participant' response and calculate the prediction accuracy for each structure among the rest of the participants, we have 0.91 for square, 0.76 for ring, 0.36 for hexagon and 0.48 for triangle. The overall accuracy is 0.62. The accuracy are generally lower (except for the square) compared to using our old criterion where we exclude participants that get more than attention checks wrong. However, the performance is still higher than that of GPT-4.

## G  Variation on prompts

**Hexagonal and triangular grid**      In this variation, we gave a more detailed description of the hexagonal and triangular grid. Example of prompts are shown in Figure 20. A similar analysis was conducted on the same dataset as Figure 3. Given that the performance of the other models other than GPT-4 are below random guessing, we only include the result from GPT-4 here. The prediction accuracy for triangle is 0.238 (with std = 0.016) and for hexagon is 0.088 (with std = 0.029), both are still within the margin of error as the results (for triangle is 0.204 with std = 0.015 and for hexagon is 0.046 with std = 0.018) from the original prompting.

**Question**: "In a hexagonal tiling (or hexagonal tessellation), equilateral hexagonal tiles are arranged to fill 2-dimensional space with no overlaps and no gaps. "Pointy-topped" refers to the orientation of the hexagon, indicating that it has a pointed or acute angle at the top. Since equilateral hexagons have equal sides and 120 degree internal angles, each vertex has 3 directions to go along the edges. Now, consider the graph given by the vertices of two rows of a Pointy-topped hexagonal tiling. The top row contains one hexagonal tile, while the bottom row has two. Starting from a vertex, you will move between vertices along the edges of these tiles. Initially, you are positioned at the bottom left corner of the map, where you find a drilling rig. You move up by one step, where you find a carousel. You move up-right by one step, where you find a loupe. You move up by one step, where you find a gown. You move up-right by one step, where you find a printer. You move down-right by one step, where you find a black-footed ferret. You move down by one step, where you find a station wagon. You move down-left by one step, where you find a bath towel. You move up-left by one step. What will you find?"
**Answer**: "loupe."

(a) Hexagon

**Question**: "In a triangular tiling (or triangular tessellation), equilateral triangular tiles are arranged to fill 2-dimensional space with no overlaps and no gaps. Since equilateral triangles have equal sides and 60 degree internal angles, each vertex has 6 directions to go along the edges: left/right, down left/right, and up left/right. Now, consider the graph given by the vertices of two rows of a triangular tiling. The top row contains one triangular tile, the bottom row has three. Hence, there are three rows of vertices: the first row has one vertex, the second has two, and the third has three. Starting from a vertex, you will move between vertices along the edges of these tiles. Initially, you are positioned at the bottom left corner of the map, where you find a box turtle. You move right by one step, where you find a hand plane. You move up-right by one step, where you find a guacamole. You move down-right by one step, where you find a table lamp. You move left by one step. What will you find?"
**Answer**: "hand plane."

(b) Triangle

Figure 20: Example question and its answer for size-2 triangular grid and size-2 hexagonal grid with more detailed description.

**Different tree traversals**      In this variation, we used breadth-first traversal method (instead of depth-first traversal in the main text) to generate the prompt, which gives a layer-wise description of the tree structure. An example of the prompt are shown in Figure 21. The prediction accuracy under both breadth-first traversal and depth-first traversal method is in Table 5. We can see they are similar (within margin of error) across different LLMs.

| | GPT-3.5 | GPT-4 | Llama-2-70b-chat-hf | CodeLlama-34b-Instruct-hf |
|---|---|---|---|---|
| depth-first traversal (Original) | 0.250 (std=0.037) | 0.506 (std=0.017) | 0.350 (std=0.025) | 0.264 (std=0.029) |
| breadth-first traversal | 0.262 (std=0.032) | 0.556 (std=0.020) | 0.316 (std=0.018) | 0.264 (std=0.025) |

Table 5: Prediction accuracy under depth-first traversal vs. breadth-first traversal for tree structure. The mean accuracy and standard deviation are calculated over 5 runs on the same dataset.

## H  Prompting CodeLlama to write code

For CodeLlama-34b, we performed an additional experiment using Program-of-Thought (PoT) prompting, where we prompt it to write a python code. An example is shown in Figure 23.

**Question**: "You have been given a tree structure with 9 nodes. The root node is a great white shark. The great white shark has 2 children: a garter snake and a Gila monster. The garter snake has 2 children: a jigsaw puzzle and a moped. The Gila monster has 2 children: a Christmas stocking and a horse-drawn vehicle. The jigsaw puzzle has a child: a Tibetan Terrier. The moped has a child: an umbrella. The Christmas stocking has no children. The horse-drawn vehicle has no children. The Tibetan Terrier has no children. The umbrella has no children. What is the cousin of the moped? "

**Answer**: "Christmas stocking, horse-drawn vehicle."

(a) Tree (Breadth-First Traversal)

Figure 21: Example question and its answer for the tree structure under breadth-first traversal.

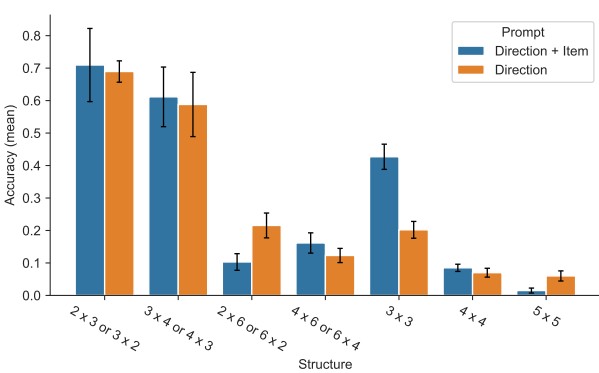

Figure 22: the pattern of "Direction + Item" > "Direction" can be observed when we group by the area size. The mean and the standard deviations are calculated across 10 different runs.

```python
def solution():
    n = 3
    square_grid = [[None for j in range(n)] for i in range(n)]
    # We use the coordinate system that is consistent with
    # OpenCV's conventions.
    # Initial position is bottom right corner of the grid so
    # the coordinate of the initial position is (2, 2)
    square_grid[2][2] = "yellow lady's slipper"
    # you move left by one step, so the new coordinate is (2, 1)
    square_grid[2][1] = "swimming cap"
    # you move left by one step, so the new coordinate is (2, 0)
    square_grid[2][0] = "bell pepper"
    # you move up by one step, so the new coorindate is (1, 0)
    square_grid[1][0] = "poke bonnet"
    # you move up by one step, so the new coordinate is (0, 0)
    square_grid[0][0] = "leaf beetle"
    # you move right by one step, so the new coordinate is (0, 1)
    square_grid[0][1] = "lion"
    # you move right by one step, so the new coordinate is (0, 2)
    square_grid[0][2] = "Chow Chow"
    # you move down by one step, so the new coordinate is (1, 2)
    square_grid[1][2] = "plate"
    # you move left by one step, so the new coordinate is (1, 1)
    square_grid[1][1] = "tow truck"
    # you move down by one step, so the new coordinate is (2, 1)
    answer = square_grid[2][1]
    return answer
```

Figure 23: An example of code prompting for CodeLlama.

We tested CodeLlama-34b and CodeLlama-13b (which is more tailored for autocompletion) for the square grid of size 3:

|  | CodeLlama-13b (PoT) | CodeLlama-34b (PoT) | CodeLlama-34b-Instruct-hf (No PoT) |
|---|---|---|---|
| Accuracy | 0.25 | 0.20 | 0.25 |

Table 6: Prediction accuracy of CodeLlama models with code prompting.

As we can see in Table 6, there wasn't any improvement in accuracy. Upon examining the outputs, we noticed that the generated code often suggest incorrect navigation, resulting in wrong answers. This implies that while using code prompts works well for simple arithmetic, it struggles with spatial navigation tasks. Additionally, while this method might be suitable for simple grids like squares, it's not yet clear how to effectively apply it to more complex grid patterns, such as those involving hexagons and triangles.

