# OpenReview forum: "Evaluating Spatial Understanding of Large Language Models"
_TMLR — Accepted by TMLR_

### Review · Reviewer_VpRX · 2023-10-22

**Summary Of Contributions:**

This work investigates the spatial reasoning abilities of LLMs, exploring a number of factors, including topology (ring, grid, hexagon, etc), presentation format, and model size. The results indicate some degree of spatial reasoning, particularly in GPT-4, with stronger performance for rectangular grids vs. other topologies. A direct comparison is also performed with human participants.

**Audience:**

Yes

**Broader Impact Concerns:**

There are no discernible ethical implications of this work.

**Claims And Evidence:**

Yes

**Requested Changes:**

I have indicated some requested changes in the previous section.

**Strengths And Weaknesses:**

## Strengths:
- The analysis is very comprehensive, considering many factors such as model size, presentation format, and topology.
- The error analysis provides interesting insight into the models' underlying representations.
- A comparison with human behavior is performed, revealing a similar trend of difficulty with respect to topology (though overall better performance in humans vs. LLMs).

## Weaknesses:
- I found the description of the non-rectangular grids to be somewhat confusing (e.g. I had to think quite a bit to visualize a 'pointy-topped regular hexagonal tile map'). I think this may explain the relative difficulty of the other types of grids. This is also consistent with the fact that human participants had higher error rates on these grids. It may be worth mentioning that errors on these grids may be related to the inherent difficulty of describing them with natural language (as opposed to visually).
- Related to the previous point, there is a concern that the non-rectangular attention checks are not easy enough to be considered attention checks. How do the human results look when participants are only excluded for getting the rectangular attention check wrong?
- Even though the human performance is better than GPT-4, there is a very strong correspondence in the relative performance across different topologies. This strongly suggests that some common factor may explain the relative difficulty of these different topologies in both human participants and LLMs. This should be emphasized in the text.
- Can a more detailed description of the human participants and study procedure be provided? e.g., how were they recruited? What was the age range of the participants? how was the study performed (online? in-person?)?
- Some measure of error (e.g. standard error) should be included for all results, including tables and figures (as error bars).
- How were the models evaluated (e.g. through the OpenAI API, or the ChatGPT web interface?)? It would be helpful to provide more details about the model settings (e.g. temperature, top p), if this information is available.

## Minor comments:
- The citations should use \citep unless the name is explicitly mentioned in the sentence.
- In the introduction, the statement about humans using spatial representations for abstract tasks should cite the following paper:
Constantinescu, A. O., O’Reilly, J. X., & Behrens, T. E. (2016). Organizing conceptual knowledge in humans with a gridlike code. Science, 352(6292), 1464-1468.
- In figure 3, colors are used to indicate model, but in figure 4 colors are used to indicate the task. I would suggest to use a consistent scheme throughout the entire paper (e.g., colors to represent model for all figures). Similarly, I would suggest to keep the models in a consistent order.
- In describing the relationship between the present work and Momennejad et al. (2023), it should be more clearly noted that these two works primarily investigate different (but related) tasks, e.g. path integration for sensory prediction in the present work, vs. planning (generating a path to move from point A to point B) in Momennejad et al. (2023).
- In the example given for the chain-of-thought prompt, the answer is stated before the chain-of-thought. Just to clarify, is the model then allowed to generate the chain-of-thought before producing an answer?

---

> ### Author Response · Authors · 2023-12-22
> **response**
>
> We thank the reviewer for the detailed and constructive review. We are glad to learn that the reviewer finds our analysis comprehensive and interesting. Below is the response to the weakness and minor comments the reviewer raised.
>
> **Concern for non-rectangular grid**
> We agree with the reviewer that the fact that both human participants and LLMs had higher error rates on the non-rectangular grids can be due to the fact that these grids are inherently difficult to be described in natural language. Another common factor (as we mentioned in Section 3.1) is that the non-rectangular grids are much less encountered by both humans and LLMs compared to the square grid, where there is more prevalence of tabular data and city grid navigation within the model’s training data. We've added a paragraph in Section 5 of our revised manuscript to emphasize these factors.
>
> As the reviewer suggests, we look into the experiment results if we only exclude the participants that get the rectangular attention check wrong. Among the 23 participants, only one participant failed the attention check for the rectangular grid, and this participant got 3 out of 4 attention checks wrong. If we exclude this participant’s response and calculate the prediction accuracy for each structure among the rest of the participants, we have 0.91 for square, 0.76 for ring, 0.36 for hexagon, and 0.48 for triangle. The overall accuracy is 0.62. The accuracies are generally lower (except for the square) compared to using our old criterion where we exclude participants that get more than attention checks wrong. However, the performance is still higher than that of GPT-4. We've added this result in the Appendix.
>
> **Recruitment of participants**
> We recruited participants online through an online crowdsourcing platform (Prolific.com). It is a platform for researchers to post behavioral studies and recruit participants. The way it works is that we first create a web interface for our study and upload it to Prolific with estimated finish time, hourly payment, and participants' criteria, then the platform will send out the study invitation to participants who match the criteria. Participants take part in the study online and remotely using their own computers.
>
> **Impact of the exclusion criteria**
> In our experiment, we set the participants' criteria to be fluent in English as answering the prompts required an understanding of English, and the study to be evenly distributed to all available participants. The demographic data of the 23 participants that took part in our study is as follows: age range (min: 21, max: 62, median: 36, mean: 41.9), gender (Male: 12, Female: 10, No data: 1), employment status (Full-time: 9, Not-in-paid-work: 5, part-time:3, unemployed: 2, No data: 4), student status (Yes: 3, No: 17, No data: 3). In the experiment, we seek to mitigate the bias by setting as few criteria as possible, but it is inevitably skewed to people with computers and internet connection, and some characteristics of the WEIRD (Western, Educated, Industrial, Rich, Democracies) population, which is unfortunately still a common practice across much of cognitive science and psychology research. We've added these results in the Appendix.
>
>
> **Measure of error**
> For all the experiments presented in the main paper, we conducted each experiment five times to calculate and report the mean and standard deviation. We’ve updated all the figures with error bars and tables with standard errors in our revised manuscript.
>
> **Details about the model settings**
>
> The models were evaluated using OpenAI API, and we used the following decoding parameters:
> ```
> Frequency_penalty = 0.0
> Presence_penalty = 0.0
> Temperature = 1.0
> Top_p = 1.0
> ```
>
> **Minor comments**
> We fixed the \citep issue and cited the paper in the introduction section as suggested. We also updated the color and model order in the figures for better coherence. We added a paragraph to better discuss the relationship between the present work and Momennejad et al. (2023) in Section 6 of our revised manuscript.
>
> > In the example given for the chain-of-thought prompt, the answer is stated before the chain-of-thought. Just to clarify, is the model then allowed to generate the chain-of-thought before producing an answer?
>
> Yes, the chain-of-thought is generated before producing the final answer in our experiments. We changed the order of 'Answer' and 'CoT' in the example figure (Fig 16) to clarify this.

---

> > ### Comment · Reviewer_VpRX · 2023-12-26
> > **Followup**
> >
> > Thank you very much to the authors for this thorough response. All of my concerns have been addressed.

---

### Review · Reviewer_zsuW · 2023-10-31

**Summary Of Contributions:**

This paper aims to evaluate LLM's understanding of spatial structures, including square, hexagonal, triangular grids, rings and trees. The authors propose a suite of navigation tasks rendered in natural language, which critically require understanding of spatial structures to correctly reason about the underlying maze. Experiments show that only GPT4 and the largest LLaMA models (LLaMA2-70b & CodeLLaMA-34b) achieve non-trivial performance, with success rate biasing towards squares compared to other topologies.

**Audience:**

Yes

**Broader Impact Concerns:**

No ethical concerns

**Claims And Evidence:**

No

**Requested Changes:**

**Crucial**
- Report decoding hyperparameters. It is also highly recommended to try different decoding techniques or aggregating results from multiple decoding runs to report consistency.
- Adjust the prompt to see if models were performing badly due to the difficulty in understanding the task vs. performing the task.
- Allow the models to use scratchpads and allow the code model to use programs to aid their work.
- Clarify motivation for the size-inference experiment
- Re-evaluate human performance or explain the human errors when their performance is far from perfect (e.g. 0.41 for Hexagon and 0.58 for Triangle).

**Strengths And Weaknesses:**

**Strengths**
- A novel task setup that converts the evaluation of spatial understanding into measuring navigation accuracy.
- An important empirical finding that particular patterns in the training data have made LLMs more familiar with squares than other topologies.

**Weaknesses**
- The task setup is interesting but may be conflated by other variables.

For example, models may have difficulties due to either the unnatural language prompt (e.g. Figure 5 prompt for tree) or the lack of scratchpad space to workout their solutions.
Specifically, the prompt for tree structure implicitly requires the model to translate a layer-wise traversal result to a tree structure. I'd suggest exploring other traversal ways, or tree representations. The lack of scratchpad space may introduce extra difficulty orthogonal to spatial understanding, as humans would also find it hard to construct a spatial topology and keep track of the object on each node solely by imagination. Besides, CodeLLaMA might be better at writing codes to construct the tree structure so program-of-thought (PoT) prompting is suggested to reveal a more comprehensive picture on this problem.
- No decoding hyperparameters (e.g. temperature, max_tokens, number of runs) reported. It is a common practice to try different decoding techniques (e.g. top-p, top-k, majority vote).
- Unclear motivation of the size-inference experiment (Section 3.5). I'm having a hard time understanding how this experiment connects to the previous navigation experiment. Is it a pre-requisite for successful navigation on a maze with a certain shape? What would success or failure on size-inference imply about navigation ability, or the more fundamental spatial understanding ability? What does the finding of better performance with item information than without suggest? Is it possible that item information has actually made size-inference easier by counting unique object names?
- The human baseline is surprisingly low. I expected humans to find this task trivial but the reported human performance is unexpectedly low (e.g. 0.41 for Hexagon and 0.58 for Triangle). I suspect that human participants did not devote enough effort but I might also missed something. Are there particular areas where humans find this task hard?

---

> ### Author Response · Authors · 2023-12-22
> **response 1**
>
> We thank the reviewer for all the detailed comments, they were very helpful during our revision process. We will address the concerns in the following points.
>
> **Decoding hyperparamers**
>
> Thank you for raising this issue. The decoding parameters have been added to the paper. They are listed below.
>
> ```
> decoding parameters:
> Frequency_penalty = 0.0
> Presence_penalty = 0.0
> Temperature = 1.0
> Top_p = 1.0
> ```
>
> Additionally, for all the experiments presented in the main paper, we conducted each experiment five times to calculate and report the mean and standard deviation. We have updated all the figures and tables in the manuscript to reflect this.
>
> ---
>
> **Variations on prompts**
>
> Thank you for the suggestion. While we focused primarily on simple prompting methods in order to investigate the fundamental spatial reasoning abilities of language models, we agree that trying different variations on the prompts is important to understanding whether the difficulty arises from understanding the task or in performing the task. We additionally experimented with a variation of our prompt for triangles and hexagons. In this variation, we gave a more detailed description of the structure, with the aim to make the structure as clear and unambiguous as possible. Example prompts are as below:
>
> *Triangle*:
> "In a triangular tiling (or triangular tessellation), equilateral triangular tiles are arranged to fill 2-dimensional space with no overlaps and no gaps. Since equilateral triangles have equal sides and 60 degree internal angles, each vertex has 6 directions to go along the edges: left/right, down left/right, and up left/right. Now, consider the graph given by the vertices of three rows of a triangular tiling. The top row contains one triangular tile, the middle row has three, and the bottom row has five. Hence, there are four rows of vertices: the first row has one vertex, the second has two, the third has three, and the fourth has four. Starting from a vertex, you will move between vertices along the edges of these tiles. Initially, you are positioned at the bottom left corner of the map, where you find a box turtle. You move right by one step, where you find a hand plane. [...] You move left by one step. What will you find?"
>
> *Hexagon*:
> "In a hexagonal tiling (or hexagonal tessellation), equilateral hexagonal tiles are arranged to fill 2-dimensional space with no overlaps and no gaps. “Pointy-topped" refers to the orientation of the hexagon, indicating that it has a pointed or acute angle at the top. Since equilateral hexagons have equal sides and 120 degree internal angles, each vertex has 3 directions to go along the edges. Now, consider the graph given by the vertices of two rows of a Pointy-topped hexagonal tiling. The top row contains one hexagonal tile, while the bottom row has two. Starting from a vertex, you will move between vertices along the edges of these tiles. Initially, you are positioned at the top corner of the map, where you find an ice pop. You move down-right by one step, where you find a Boxer.  You move down by one step, where you find apoke bonnet. [...] You move up-right by one step. What will you find?"
>
> | GPT-4  | Original  | Detailed  |
> |---|---|---|
> | Triangle  | 0.204 (std=0.015)  | 0.238 (std=0.016)   |
> | Hexagon  | 0.046 (std=0.018)  | 0.088 (std=0.029) |
>
> The mean accuracy and standard deviation are calculated over 5 runs on the same dataset.
> As shown above, we see a slight improvement but the overall pattern (e.g. squares >> rings ~ triangles > hexagons) remains the same.
>
>
> We also explored whether breadth-first traversal (BFT) affects the prediction performance on the tree structure compared to depth-first traversal (DFT), which is what we initially did in the paper. The accuracy with its standard deviation are follows:
>
> |   | GPT-3.5  | GPT-4  | Llama-2-70b-chat-hf | CodeLlama-34b-Instruct-hf |
> |---|---|---| --- | --- |
> | DFT (Original) | 0.250 (std=0.037)  | 0.506 (std=0.017)  | 0.350 (std=0.025) | 0.264 (std=0.029) |
> | BFT  | 0.262 (std=0.032)  | 0.556 (std=0.020) | 0.316 (std=0.018) | 0.264 (std=0.025)  |
>
> The mean accuracy and standard deviation are calculated over 5 runs on the same dataset. The prediction accuracy under DFT or BFT is similar (within the margin of error) across different LLMs. We've added these results with an example of prompts under BFT to the Appendix.
>
> ---

---

> ### Author Response · Authors · 2023-12-22
> **response 2**
>
> **Motivation of the size-inference experiment**
>
> Thank you for your feedback about the presentation of Section 3.5 on the size-inference task.
>
> Simultaneously inferring the extent of map while navigating is an important component of navigation as a cognitive task. Inferring the extent of a map is a non-trivial task which requires integrating information about the history of actions and reasoning about what it implies spatially. Indeed, humans often have a sense of the size of the places they visit, not only how to navigate from point A to point B. In particular, it would be possible to navigate between a pair of points without knowing the size of the map you are in. Neither task subsumes the other. But humans can do both. In Section 3.5, we evaluate whether LLMs can infer the size of the map by reasoning about the sequence of actions and their spatial implications. We have revised this section and added an expanded description of motivation behind this task.
>
>
> > What does the finding of better performance with item information than without suggest? Is it possible that item information has actually made size-inference easier by counting unique object names?
>
> While analyzing the data to compute the standard deviation from various trials, we found that there are significant variations in accuracy based on the shape of the rectangles, specifically whether they were 'tall' or 'wide'. (e.g. "3x2" is consistently better than "2x3" for "Direction+Item", while "2x3" is consistently better than "3x2" for "Direction".)
> Although the pattern (i.e. the accuracy for "Direction + Item" is higher than "Direction") can still be observed across 10 different runs when we group by the area size (see the Figure 22 in Appendix), when we distinguish 'tall' or 'fat' rectangles, the pattern is not as consistent as we originally observed.
> To reflect this observation, we removed the following statement from the abstract: "We also discover that, similar to humans, LLMs utilize object names as landmarks for maintaining spatial maps" and changed the caption of Figure 9.
>
>
> ---

---

> ### Author Response · Authors · 2023-12-22
> **response 3**
>
> **Use of scratch pads and code**
>
> We have provided the "scratch pads" approach (which we refer to as the chain-of-thought experiment in the Appendix Figure 17), where we see that CoT improves the baseline, but the performance plateaus after using 3-shots.
> We also observe that the chain-of-thought prompting performance plateaus as we increase the size of the gird (as shown in the Appendix Figure 18).
> We believe these experiments suggest the limit of the chain-of-thought prompting approach in the spatial reasoning tasks we investigated.
>
> For CodeLlama-34b, we performed an additional experiment using Program-of-Thought prompting, where we prompted it to write a Python code.
> An example is given below:
>
> ```python=
> EXAMPLE
> Question:
> You have been given a 3 by 3 square grid. At the start, you are positioned at the bottom right corner of the grid, where you find a yellow lady's slipper. You move left by one step, where you find a swimming cap. You move left by one step, where you find a bell pepper. You move up by one step, where you find a poke bonnet. You move up by one step, where you find a leaf beetle. You move right by one step, where you find a lion. You move right by one step, where you find a Chow Chow. You move down by one step, where you find a plate. You move left by one step, where you find a tow truck. You move down by one step. What will you find?
>
> Answer:
> def solution():
> 	n = 3
> 	square_grid = [[None for j in range(n)] for i in range(n)]
> 	# We use the coordinate system that is consistent with
> 	# OpenCV's conventions.
> 	# Initial position is bottom right corner of the grid so
> 	# the coordinate of the initial position is (2, 2)
> 	square_grid[2][2] = "yellow lady's slipper"
> 	# you move left by one step, so the new coordinate is (2, 1)
> 	square_grid[2][1] = "swimming cap"
> 	# you move left by one step, so the new coordinate is (2, 0)
> 	square_grid[2][0] = "bell pepper"
> 	# you move up by one step, so the new coorindate is (1, 0)
> 	square_grid[1][0] = "poke bonnet"
> 	# you move up by one step, so the new coordinate is (0, 0)
> 	square_grid[0][0] = "leaf beetle"
> 	# you move right by one step, so the new coordinate is (0, 1)
> 	square_grid[0][1] = "lion"
> 	# you move right by one step, so the new coordinate is (0, 2)
> 	square_grid[0][2] = "Chow Chow"
> 	# you move down by one step, so the new coordinate is (1, 2)
> 	square_grid[1][2] = "plate"
> 	# you move left by one step, so the new coordinate is (1, 1)
> 	square_grid[1][1] = "tow truck"
> 	# you move down by one step, so the new coordinate is (2, 1)
> 	answer = square_grid[2][1]
> 	return answer
>
> Question:
> You have been given a 3 by 3 square grid. Starting from a vertex, you will move along the edges of the grid. Initially, you are positioned at the bottom left corner of the grid, where you find a sulphur-crested cockatoo. You move right by one step, where you find a plow. You move right by one step, where you find a Malinois. You move up by one step, where you find a barbershop. You move left by one step, where you find a washing machine. You move left by one step, where you find a mongoose. You move up by one step, where you find a sarong. You move right by one step, where you find an Afghan Hound. You move down by one step. What will you find?
> ```
>
> We tested CodeLlama-34b and CodeLlama-13b (more tailored for autocompletion) for the square grid of size 3. We experimented with CodeLlama-34b-Instruct-hf as well but it was heavily fine-tuned for chatting, and could not output consistent code that follows our code prompt.
>
>
> |   | CodeLlama-13b (PoT)  | CodeLlama-34b (PoT)  | CodeLlama-34b-Instruct-hf (No PoT) |
> |---|---|---| --- |
> | Accuracy | 0.25  | 0.20  | 0.25 |
>
>
> As we can see, there wasn't any improvement in accuracy.
> Upon examining the outputs, we noticed that the generated code often suggests incorrect navigation, resulting in wrong answers. This implies that although using code prompts works well for simple arithmetic problems like GSM8K, it struggles with spatial navigation tasks. Additionally, while PoT prompting might be suitable for simple grids like squares, it's not yet clear how to effectively apply it to more complex grid patterns, such as those involving hexagons and triangles.
>
>
> ---

---

> > ### Author Response · Authors · 2023-12-22
> > **response 4**
> >
> > **Discussion of human performance**
> >
> > Thanks for your question. We believe the human errors for hexagonal and triangular grids can be attributable to two factors:
> > 1) Hexagonal and triangular grids are intrinsically harder to describe in natural language. For example, "pointy-topped" and "flat-topped" hexagonal grids are the terminology some mathematicians use to describe the two variations of hexagonal grids, but it would not be familiar to non-experts. Squares and rings are much easier to visualize and describe through natural language.
> > 2) Hexagonal and triangular grids are much less encountered in daily life, so participants might have been less confident answering questions about these structures compared to square and ring.
> >
> > We've added a paragraph emphasizing these factors in the analysis of human experiments in Section 5.

---

> > > ### Comment · Reviewer_zsuW · 2023-12-31
> > > **The authors have addressed all my requested changes.**
> > >
> > > Thank you very much to the authors for the detailed response to all my requested changes.

---

### Review · Reviewer_w7Kr · 2023-11-30

**Summary Of Contributions:**

The authors propose a wide variety of simple navigation tasks to understand the ability of existing LLMs to move across several structures. They focus on geometrically shaped systems, asking the model to tell what object is at a particular location after making a traversal of the spatial map. They very the topology of the "rooms" in the map, the way the "rooms" are laid out in the map, the way the "rooms" are laid out in the prompt, and consider the inclusion of a global map. Through all of this inference, the authors suggest that current LLMs do better in some kinds of rooms, benefit from the utilization of additional non-navigational information to reason about the path, and that LLMs generally seem to do better with local map representations as opposed to global map representations.

**Audience:**

Yes

**Broader Impact Concerns:**

I'm not entirely sure if the existing broader impact statement adequately discusses the potential social impact of the authors' work. Given that the authors did perform experiments with a human baseline, I'd like some information about the disclosure process and the steps the authors took to mitigate selection bias. I'm not entirely sure how their existing statement is focused in on the precise work that they have done.

**Claims And Evidence:**

Yes

**Requested Changes:**

- "using the rhombus grid, which was achieved by rotating the square grid 90 degrees" - did you mean 45 degrees? Rhombuses are normally not axis-aligned, and so I'd imagine this is a typo ( not critical for acceptance, but seems like an easy fix ).
 - The authors must include more detailed information about the nature of their decoding process and what precise model checkpoints they used. Given the sheer number of papers that evaluate an LLM's capability to perform some task, this is critical to ensure that the paper is taken seriously and that others can extend the work accordingly.
- The authors must also give more information about the nature of what "steps" mean at the various parts of their work, and give examples of the "random and snake" prompt orderings, even if it's in the appendix. I had issues understanding what precisely was going on in these sections.
- The authors should explain their particular choice of "steps" for their analysis, and why they included a "steps" parameter in their equation when they seem to incidate that this is a constant. Given that they focus on this parameter for their subsequent analysis, clarifying this is critical for acceptance as otherwise their claims do not seem to stand up to their analysis.
- Besides the above critical changes, I'd like some better grounding to strengthen why these tasks would be important for our understanding of how LLM's reason spatially. Some connection to existing work like SayNav would be useful, as it would allow us to better appreciate the synthetic task and the more fine-grained analysis it offers.

**Strengths And Weaknesses:**

Overall, the paper provides a fairly comprehensive comparison amongst the factors it considers. The authors succeed at demonstrating that, for the models they work with, the model is likely to benefit in navigating in a square room and that some ways of representing global representations seem to hurt performance compared to the local representations they consider.

However, there are quite a few methodological quirks that make this work hard to replicate and hard to see the support of the work from the evidence that they provide.

Firstly, and most importantly, the authors do not indicate the checkpoint of the models they are using, along with their decoding parameters for their models. For models like GPT4 and GPT3.5, it is best practice to use a model like the 1106 version so that experiments can be replicated. Additionally, it is usually highly relevant to discuss what the temperature is for these evaluative tasks, as otherwise it is difficult to confirm that the results are deterministic and also to aid in replication and further research.

Secondly, it would benefit the authors to be clearer about their design choices, and what certain terms mean. In section 3.1, the authors say to "keep the number of steps to be 8", but it is unclear from context what the "steps" mean here. Later on, the authors also use the term "exploration steps", which I am unable to find a clear definition of. Additionally, it's not clear why the authors did not use zero-shot chain of thought prompting, or any other existing strategy that is not few-shot, and if they had considered other variations of the prompt they propose. Given that even prompt formatting can influence performance, I'd like some clarification on what other designs were considered and why the authors ended up on the set that they did. In general, the paper should spend more time discussing the precise generation approach to allow for replication and wider contribution to the community.

Furthermore, the authors claim that the number of steps was statistically significant, but earlier state that they made the number of steps a constant. I do not understand how the number of steps can causally "contribute" to the error when it remains at a constant, and am unsure if the methodology is poorly explained or if the evaluation is poorly explained.

Lastly, I generally had troubles understanding why the authors included several models besides GPT-3.5 and GPT-4. Given that there is minimal analysis of any model besides GPT-3.5 and GPT-4, it's confusing as to what the benefit of this analysis really is, and what I should take away from this analysis other than LLaMA generally failing at these tasks.

---

> ### Author Response · Authors · 2023-12-22
> **response 1**
>
> We thank the reviewer for their detailed and thoughtful comments. We detail our responses below.
>
> **Details of model checkpoints and decoding parameters**
>
> Thank you for raising this issue. The decoding parameters have been added to the paper. They are listed below.
>
> ```
> decoding parameters:
> Frequency_penalty = 0.0
> Presence_penalty = 0.0
> Temperature = 1.0
> Top_p = 1.0
> ```
>
> The model checkpoints are listed in Section 2.1. We test GPT-3.5 (gpt-3.5-turbo-0301), GPT-4 (gpt-4-0314), Llama2-7B, Llama2-13B, Llama2-70B, and CodeLlama-34B.
>
> ---
>
> **Clarification on the number of "steps" in Section 3.1**
>
> Thank you for your feedback about the clarity of the presentation in Section 3.1.  We have made revisions to the paper to make this clearer. We would like to explain here as well.
>
> “Step” in the regression refers to the number of *movements* traversed on the graph from the start to the goal in the prompt. For example, in the example prompt for the 2 by 2 grid in Figure 2, the number of steps is 3 because “you” are asked to move right by one step, move up by one step and move left by one step consecutively. We've added a definition of the term "navigation step" and replaced all "step" with "navigation step" for better clarification in the revised manuscript.
>
> In the logistic regression, the number of navigation steps is not fixed and is part of the input to the model. Thus, the logistic regression models the difficulty of a problem instance (via its error rate) as a function of the structure *and* the number of navigation steps. The logistic regression is fitted on different structures and different numbers of navigation steps simultaneously.
>
> In Figure 3, we compare the accuracy of the models across the different spatial structures. We condition on a fixed number of navigation steps (which is 8) in order to get a measure of the difficulty of each structure, independently of the number of navigation steps taken on the graph. That is, the number of navigation steps is the same, but the graph structure is different, which allows us to more fairly attribute differences in error rate to the structure itself. We do this because, as indicated by the logistic regression, a larger number of navigation steps increases the error rate for all structures. Conditioning on the number of navigation steps helps ensure a fairer comparison between different structures.
>
> We’ve changed the order of the logistic regression and the descriptions of Figure 3 to avoid confusion and added a paragraph in the revised manuscript to better clarify this.

---

> ### Author Response · Authors · 2023-12-22
> **response 2**
>
> ---
>
> **Variations on prompts**
>
> > Additionally, it's not clear why the authors did not use zero-shot chain of thought prompting, or any other existing strategy that is not few-shot, and if they had considered other variations of the prompt they propose. Given that even prompt formatting can influence performance, I'd like some clarification on what other designs were considered and why the authors ended up on the set that they did. In general, the paper should spend more time discussing the precise generation approach to allow for replication and wider contribution to the community.
>
> Thank you for the question. When designing and choosing the prompts, we focused primarily on simple prompting methods in order to investigate the fundamental spatial reasoning abilities of language models. By starting with these basic prompts, we hope to establish a solid baseline for their capabilities before delving into more complex techniques, as the area of spatial reasoning in LLMs is still under-explored.
>
> Specifically, regarding the prompts for hexagons and triangles, given that these structures are intrinsically hard to describe in natural language, we aimed to generate the prompts as clearly and unambiguously as possible. For example, in our prompt, by using the word "pointy-topped" hexagonal grid, we hope to distinguish its orientation from the "flat-topped" hexagonal grid.
>
> Here, we additionally experimented with a variation of our prompt for triangles and hexagons, providing a much more detailed description of the structure. An example prompt for hexagons and triangles is below:
>
> *Triangle*:
> "In a triangular tiling (or triangular tessellation), equilateral triangular tiles are arranged to fill 2-dimensional space with no overlaps and no gaps. Since equilateral triangles have equal sides and 60 degree internal angles, each vertex has 6 directions to go along the edges: left/right, down left/right, and up left/right. Now, consider the graph given by the vertices of three rows of a triangular tiling. The top row contains one triangular tile, the middle row has three, and the bottom row has five. Hence, there are four rows of vertices: the first row has one vertex, the second has two, the third has three, and the fourth has four. Starting from a vertex, you will move between vertices along the edges of these tiles. Initially, you are positioned at the bottom left corner of the map, where you find a box turtle. You move right by one step, where you find a hand plane. [...] You move left by one step. What will you find?"
>
> *Hexagon*:
> "In a hexagonal tiling (or hexagonal tessellation), equilateral hexagonal tiles are arranged to fill 2-dimensional space with no overlaps and no gaps. “Pointy-topped" refers to the orientation of the hexagon, indicating that it has a pointed or acute angle at the top. Since equilateral hexagons have equal sides and 120 degree internal angles, each vertex has 3 directions to go along the edges. Now, consider the graph given by the vertices of two rows of a Pointy-topped hexagonal tiling. The top row contains one hexagonal tile, while the bottom row has two. Starting from a vertex, you will move between vertices along the edges of these tiles. Initially, you are positioned at the top corner of the map, where you find an ice pop. You move down-right by one step, where you find a Boxer.  You move down by one step, where you find apoke bonnet. [...] You move up-right by one step. What will you find?"
>
> | GPT-4  | Original  | Detailed  |
> |---|---|---|
> | Triangle  | 0.204 (std=0.015)  | 0.238 (std=0.016)   |
> | Hexagon  | 0.046 (std=0.018)  | 0.088 (std=0.029) |
>
> The mean accuracy and standard deviation are calculated over 5 runs on our dataset.
> As shown above, we see a slight improvement but the overall pattern (e.g. squares >> rings ~ triangles > hexagons) remains the same. We have added this result to the Appendix.

---

> > ### Author Response · Authors · 2023-12-22
> > **response 3**
> >
> > **Evaluation of models other than GPT-3.5 and GPT-4**
> >
> > We evaluate error rates for GPT-3.5, GPT-4, and LLaMa. This evaluation assesses the relative ability of different models at these spatial understanding tasks. GPT-3.5 and the LLaMa family of models had high error rates on most tasks, with the exception of the square grid. Due to these high error rates (not better than random guessing), an error analysis would not reveal much, as these models demonstrate little to no understanding of these structures. However, GPT-4 has better-than-random performance, demonstrating *some* "understanding", and hence it would be interesting to perform an error analysis which reveals any patterns in the modes of failure.
> >
> > ---
> > **Requested changes**
> > > "using the rhombus...
> >
> > The degree of rotation is indeed 45, not 90. Thank you for pointing it out, we've fixed it in the revised manuscript.
> >
> > > The authors must include more detailed information about the nature of their decoding process and what precise model checkpoints they used.
> >
> > A description of the model checkpoints and decoding parameters has been added to the manuscript.
> >
> > > The authors must also give more information about the nature of what "steps" mean...
> >
> > We've made changes to Section 3.1 to better explain what we mean by "steps" and added an example of "random" and "snake" order prompts in Figure 6.
> >
> > > The authors should explain their particular choice of "steps" for their analysis...
> >
> > This change has also been integrated into Section 3.1 (please also see the response above).
> >
> >
> > > Besides the above critical changes, I'd like some better grounding to strengthen why these tasks would be important for our understanding of how LLM's reason spatially. Some connection to existing work like SayNav would be useful, as it would allow us to better appreciate the synthetic task and the more fine-grained analysis it offers.
> >
> > Thank you for suggesting the connection to embodied agents and additional reference. We've added a paragraph in Section 6 to reflect this.
> >
> >
> >
> > ---
> > **Disclosure process in human experiment**
> > Thank you for your question.  Before the experiment starts, all participants are provided with an online form with information about the experiment, including the purpose, a brief description, the procedure, confidentiality, and voluntary participation. The participants are informed that their responses will be confidential and they are free to decline or end participation at any time. In the “agreement to participate” section, we need participants to confirm that by clicking the “start” button, they acknowledge that they have read the above information, and agree to participate in the study. The experiment only starts after they click the “start” button.
> >
> > The participants were recruited from an online crowdsourcing platform (Prolific.com), it is a platform for researchers to post behavioral studies and recruit participants. We set the only participants' criteria to be “fluent in English” as all our prompts and questions are written in English. Our study is evenly distributed to all available participants. The demographic data of the 23 participants that took part in our study is as follows: age range (min: 21, max: 62, median: 36, mean: 41.9), gender (Male: 12, Female: 10, No data: 1), employment status (Full-time: 9, Not-in-paid-work: 5, part-time: 3, unemployed: 2, No data: 4). Although we seek to mitigate the bias by setting as few criteria as possible, there is still a bias toward English-speaker with computers and internet connection, as well as some characteristics of the WEIRD (Western, Educated, Industrial, Rich, Democracies) population. Unfortunately, this is presently common practice across much of cognitive science and psychology research.
> >
> > We have added these paragraphs in the Broader Impact Statement.

---

> > > ### Comment · Reviewer_w7Kr · 2024-01-07
> > > **The authors have accomodated all of my changes**
> > >
> > > I'd like to thank the authors for updating the paper accordingly.

---

### Decision · Action_Editor_TPJV · 2024-01-17

**Recommendation:** Accept as is

**Comment:**

All reviewers are happy with the experiments produced and the responses/changes (including to the rhetoric of claims).

**Audience:**

Growing field of interest in LLM based reasoning, and particularly relevant to the growing space of LLM based planning for embodied agents.

**Claims And Evidence:**

The work aims to evaluate how well language models can reason about spatial relations by setting up synthetic navigation tasks where steps are described in language about differently shaped layouts/rooms (based on basic geometry). Variously sized models and humans are evaluated across the conditions, noting both the difficulty effects of size and varying shape.